# Molecular Delivery of Cytotoxic Agents via Integrin Activation

**DOI:** 10.3390/cancers13020299

**Published:** 2021-01-15

**Authors:** Martina Cirillo, Daria Giacomini

**Affiliations:** Department of Chemistry “Giacomo Ciamician”, Alma Mater Studiorum University of Bologna, Via Selmi 2, 40126 Bologna, Italy; martina.cirillo3@unibo.it

**Keywords:** integrins, active targeting, molecular delivery, receptor targeting, cancer, selective citotoxycity

## Abstract

**Simple Summary:**

Chemotherapy constituted a cornerstone in cancer treatment, but it suffers from non-selective drug cytotoxicity; an active drug targeting exerted by covalent conjugates between antitumor agents and some integrin ligands should have a more selective antitumor efficacy and could be an effective answer to reduce intolerable side-effects of current therapies. In this review we focused on those cytotoxic agents that were covalently inserted in molecular cargos characterized by specific integrin ligands. It was demonstrated that the integrin recognizing fragments were able to switch on an active and selective targeting against tumor cells.

**Abstract:**

Integrins are cell adhesion receptors overexpressed in tumor cells. A direct inhibition of integrins was investigated, but the best inhibitors performed poorly in clinical trials. A gained attention towards these receptors arouse because they could be target for a selective transport of cytotoxic agents. Several active-targeting systems have been developed to use integrins as a selective cell entrance for some antitumor agents. The aim of this review paper is to report on the most recent results on covalent conjugates between integrin ligands and antitumor drugs. Cytotoxic drugs thus conjugated through specific linker to integrin ligands, mainly RGD peptides, demonstrated that the covalent conjugates were more selective against tumor cells and hopefully with fewer side effects than the free drugs.

## 1. Introduction

Great efforts have been devoted to find innovative therapies for cancer [1,2,3]. Chemotherapy constituted a cornerstone in cancer treatment but it suffers from several limitations, among others, it presents several intolerable side-effects due to non-specific targeting, and the appearance of multiple drug resistance [4]. To get over these difficulties, a preferential localization of the anticancer agents has been chosen as target strategy. Thus, specialized delivery systems and targeting approaches to get selective, effective therapeutic and diagnostic modalities have been studied [5,6]. Extensive efforts have been made to design and realize focused therapeutics in response to specific biomarkers, receptors, and tumor microenvironments for a precise therapy of cancer. These strategies are referred to as “active targeting” complementary to a passive targeting that was referred to specific pathophysiological characteristics of tumor tissues [7].

Tumor cells overexpress some receptors, and this could be used to functionalize nanotherapeutics to recognize the receptors on the tumor surface (active targeting) leading to a preferential accumulation within the tumor via receptor-mediated endocytosis [7,8,9].

Among targets, folate receptors were the most commonly selected cancer targets, particularly for ovarian cancers [10,11]; the riboflavin receptor, instead, emerged in recent years because carrier protein of vitamin B_2_, and three riboflavin transporters were found overexpressed in several cancers and could be chosen for tumor targeted drug delivery systems [12]. CD44, a cell surface adhesion receptor for hyaluronic acid (HA), is highly expressed in many cancers and regulates metastasis via recruitment to cell surface; therefore, various HA-based drug delivery systems have been developed for a CD44-mediated tumor targeting [13,14]. Enhanced expression of CD133, a cell surface antigen able to detect and isolate cancer stem cells, correlated with shorter patient lifespan and more aggressive disease, then monoclonal antibodies carrying drugs or toxins are used in targeting CD133 to enhance the immune response towards the disease [15]. The epidermal growth factor (EGF) receptor family is highly involved in triggering an uncontrolled signal transduction [16]. EphA2 receptor has much potential due to a possible correlation with tumor progression and metastatic spread through inhibition of receptor oligomerization and activation [17].

Integrins are heterodimeric transmembrane receptors constituted by two protein chains, α and β, and particularly attractive as pharmacological targets [18,19]. These cell adhesion glycoproteins have a fundamental role in increasing invasion, migration, and proliferation; moreover, they have been linked to tumor angiogenesis, which is an essential process for tumor growth and metastasis [20,21,22,23]. Changes in the expression of integrins in immune and stromal cells were described to support aggressive tumoral phenotypes [24]. The most studied integrins in oncology are the RGD-integrins such as α_v_β_3_/β_5_ and α_5_β_1_ integrins [25]. These integrins recognize the tripeptide arginine–glycine–aspartic acid (RGD) as the minimal sequence in extracellular matrix (ECM) proteins. Among non-natural ligands, particular attention was ascribed to cyclic peptides containing the RGD sequence into the cycle (cRGDs, Figure 1). Since the earliest studies, cRGDs showed higher potency than linear RGD sequences in the inhibition of integrin receptor [26]. This result could be ascribed to a lower flexibility due to conformational constrains by the cyclic structure and thus a higher directionality of the non-covalent interactions engaged by the cyclic molecules in the integrin-binding site. In a snapshot of the ligand-binding region of integrin α_v_β_3_ with cRGDfK (X-ray structure 1L5G, PDB data bank) [27] the ligand interactions with protein residues are highlighted.

The peptide inserts into a crevice on the integrin head: the ligand binding site. The arginine residue of the ligand inserts into a narrow groove of the α_v_ chain and engages salt bridges with aspartates 150 and 218, whereas the aspartate of the ligand coordinates the Mn^2+^ at the metal ion-dependent adhesion site (MIDAS) of the β_3_ chain. Interrupting the interactions between integrins and extracellular ECM through RGD-like compounds as inhibitors appeared a valuable strategy to trigger tumor cell apoptosis. The failure of Cilengitide, a cyclic RGD peptide inhibitor, in phase II/III clinical trials may be due to the complexity of integrins biology; the interest for these receptors was lower. However, recent literature pointed out to reconsider these important receptors [25,26,27,28,29,30] as target in the cancer research.

In the last few years, among the various developed delivery systems, peptides have been widely applied as carriers for receptor recognition and active-targeting of cytotoxic drugs to cancer cells [31,32]. In particular, on targeting integrin receptors, linear and cyclic RGD peptide motifs (Figure 1) have been employed as delivery agents for small molecular weight drugs, peptides, and proteins to tumor endothelial cells [33,34,35,36,37].

The aim of this review was to consider the most recent papers focusing on molecular delivery systems for specific cytotoxic drugs and rationally designed for an active targeting towards integrin receptors. We focused the attention on those antitumor drugs which were covalently included in a molecular cargo for a selective delivery to integrins by specific molecular fragments.

To highlight the integrin targeting portion, in all figures it was colored blue and only selected activity data were reported in the figures.

We chose to limit literature examination to covalent molecular conjugates (cargo) excluding antitumor drug encapsulation in micelles, liposomes, nanomaterials, or nanoparticles functionalized with integrin ligands and recently reviewed [38,39,40].

## 2. Drugs with a Covalent Binding to DNA

### 2.1. CisPlatin

Cisplatin has a crucial role in the treatment of several tumors; however, its efficacy and applicability are heavily restricted by severe systemic toxicities and by the appearance of drug resistance [41,42,43]. To circumvent these problems, it should be increased the selectivity of platinum drugs against cancer cells respect normal cells, by modification with specific carrier molecules whose receptors are overexpressed in tumor cells, such as integrins.

Lippard et al. [44] reported the direct tethering of integrin binding peptides containing RGD (Arg-Gly-Asp), NGR (Asn-Gly-Arg), or cyclic (CRGDC)c, (RGDfK)c, attached to platinum (IV) by an amide linkage through a succinate group (Figure 2). This upon intracellular reduction to cis platinum exerted improved anticancer activity to selectively target tumor cells over healthy cells. They demonstrated that integrin α_v_β_3_ mediated the anti-proliferative effect of the new Pt(IV)-RGD conjugates.

Casini et al. [45] developed the cisplatin encapsulation in Pd-cages decorated with some integrin binding ligands that could enable a better delivery approach, minimizing the systemic toxicity of the drug while reducing its uncontrolled speciation (Figure 3). Cages are chemical compounds with a special 3D structure that feature a cavity to host small drug molecules. On modifying specific residues, it would be possibly to allow a fine tuning of the overall properties of the cages [46].

The authors realized four conjugates of Pd_2_L_4_ metallacage with integrin ligands for subtype specific targeting of integrin α_v_β_3_, or α_5_β_1_. Cage complexes with the RGD–ligands featured strong binding and higher selectivity for their target integrin with affinities in the low nanomolar range in solid-phase binding assay tests. Formation of the host-guest complex with cisplatin and Pd-cage was supported by ^195^Pt NMR spectroscopy. The bioconjugated cage C2 significantly increases its cytotoxic potency against the α_v_β_3_ integrin expressing A375 cells (2-fold more potent with respect to free cisplatin) and it does not increase the cisplatin toxicity against the A549 cells with no expression of α_v_β_3_ such as A549 cells (lung).

Marchán et al. [47] investigated the tetrameric RAFT-RGD peptide **A** (Figure 4) to deliver Pt(IV) complexes selectively into cancer cells through α_v_β_3_ and α_v_β_5_ integrins, on considering the potential of multimeric RGD-fragments for tumor imaging and for a targeted drug delivery. They supposed that the higher binding affinity of the RAFT-RGD (**B**) to integrins could result in a higher accumulation of Pt(IV) in cancer cells, compared to the monomeric ligand (**A**) and, at last, in a higher cytotoxicity of **B**. In order to evaluate the biological activity of the compounds, the SK-MEL-28 malignant melanoma cell line was selected because of a high expression level of α_v_β_3_ and α_v_β_5_ integrins. The corresponding fluorescein-labeled peptides revealed a good cellular uptake and an internalization of the tetrameric RAFT-RGD **B** considerably higher than the monomeric analog **A**. Cellular uptake was instead reduced in CAPAN-1 pancreatic cancer cells and 1BR3G fibroblasts, selected as the negative control because their low expression of α_v_β_3_ and α_v_β_5_. The overall results highlighted the great capacity of RAFT-RGD to target SK-MEL-28 cells, to lead a higher intracellular accumulation of the platinum pro-drug and, consequently, a higher antitumor activity.

Interestingly, conjugation of a Pt(IV) prodrug to a cyclic RGD-peptide and photo activated with guanosine 5-monophosphate (5′-GMP) led to an Pt-GMP adduct with an increased phototoxicity in melanoma cancer cells overexpressing α_v_β_3_-integrin (Figure 5) [48]. It is worth mentioning that the cRGD-Pt prodrug showed a higher accumulation in SK-MEL-28 cells than Pt drug without cRGD, thus supporting the efficacy of the integrin ligand in supporting the intracellular drug transport.

### 2.2. Nitrogen Mustard Derivatives

Thysiadis et al. [49] studied two cRGDyK conjugates with 3-(4-(bis(2-chloroethyl)amino) phenoxy)propanoic acid (POPAM) a derivative of nitrogen mustards, an important class of alkylating drugs for cancer therapy [50]. Improvement of POPAM-cRGDyK inhibitory activity against the lung adenocarcinoma cells A549 (GI_50_ = 12 ± 1 μM; TGI = 90 μM), mammary carcinoma cells MCF7 (GI_50_ = 50 ± 2 μM), and prostate cancer cells PC3 (GI_50_ = 74 ± 7 μM) was observed respect POPAM alone, and it was directly related to integrins α_v_β_3_/α_v_β_5_ expression (Figure 6). The activity of POPAM conjugates was improved both in terms of potency and selectivity by conjugation to c(RGDyK) with the dithiol linker (POPAM-cRGDyK-S-S), the best result was obtained against A549 cells with GI_50_ = 8 ± 2 μM.

Chlorambucil induces DNA alkylation, and miscoding and inhibition of DNA replication. Chlorambucil is used in the treatment of several types of lymphomas. However, it showed serious problems such as a high systemic toxicity, low stability by rapid hydrolysis, and development of drug resistance. Gilad et al. [51] designed some new Chlorambucil derivatives by conjugation of the drug with a cyclic RGD peptide in order to target cancer trough integrins (Figure 6). The binding affinity of Chlorambucil-cRGDfX for integrin α_v_β_3_ was preserved upon conjugation of the drug. Human non-small cell lung carcinoma cell line H-1299 and murine melanoma cell line B16F10 as α_v_β_3_ integrin overexpressing cell lines together with human embryonic kidney-293 cell line (HEK-293) as α_v_β_3_ negative control were chosen to test the new compound. Interestingly, the Chlorambucil-cRGDfX conjugate showed an increased cytotoxic activity respect Chlorambucil alone with certain selectivity against H-1299 and B16F10 respect HEK-293.

## 3. Antimetabolites

### 3.1. Fluorouracil

Fluorouracil (5-FU) is one of the most important molecules to treat colorectal cancer [52,53]. It is a fluorinated pyrimidine antimetabolite, which irreversibly inhibits the enzyme thymidylate synthase and prevents cell proliferation by reducing the thymidine formation required for DNA synthesis [54]. It is widely used to treat breast, colorectal, and gastric cancers, but it showed several serious side effects such as toxic cardiac reaction, myelosuppression, mucositis, nausea, emesis, and development of resistance [55,56].

Thysiadis et al. [49] realized a 5-FU conjugate with cRGDyK analogously to POPAM presented above (Figure 7). The conjugate 5-FU-cRGDyK did not show any cytotoxic activity, whereas 5-FU-cRGDyK-S-S exhibited mild cytostatic activity against A549 cell with GI_50_ and TGI values 26 ± 1.5 μM and 80 ± 3 μM, respectively, with evidence of an early 5-FU release from the conjugate.

### 3.2. Methotrexate

Methotrexate (MTX) is an anti-metabolite of folic acid and acts as an anticancer and immunosuppressant agent [57]. Being highly ionized, MTX crosses very poorly the biological barriers and it needs an active transport system such as proton-coupled folate transporters or it uses receptor-mediated endocytosis mechanism via folate receptors. Intracellularly, MTX is metabolized to a polyglutamate derivative, MTX_Glu_, that shows significantly increased cell residence time and bioactivity in comparison to initial MTX form. MTX_Glu_ deprives a cell of precursors for the synthesis of DNA and RNA necessary for cell proliferation leading to DNA synthesis disturbances and cell apoptosis.

Kotamraj et al. [58] reported the design and synthesis of a sort of Methotrexate-prodrug consisted of MTX as a model anticancer agent, RGD as targeting moiety, and a β-hairpin peptide (Figure 8). The β-hairpin peptide was introduced to reduce a premature activation of the prodrug in the blood. The binding specificity of the prodrugs to α_V_β_3_ integrins was demonstrated by endothelial adhesion assay on modified HUVEC cells with a high expression of β_3_ integrins. The conjugate showed an increased stability to endopeptidase hydrolysis in plasma. It was then tested with the modified HUVEC cells and upon α_V_β_3_ integrins an increased release of Methotrexate occurred. Preferential binding of the MTX-RGD-conjugate is followed by a specific enzyme mediated hydrolysis results in accumulation of MTX in cancer cells overexpressing the integrin α_V_β_3_. In this study, a molecular modeling was used to support the propensity of hairpin formation of MTX-RGD-conjugate and the binding-induced unfolding of MTX-hairpin-RGD.

## 4. Cytotoxic Antibiotics

### Doxorubicin and Daunorubicin

Doxorubicin (DOX) is a broadly used and effective anticancer agent but it is cause of cardiotoxicity [59]. In chemotherapy, it has been considered the gold standard in oncology and often adopted as a model for the development of drug conjugates. Due to the chromophoric anthracycline group, DOX could be used for imaging studies providing information on drug distribution in tissues. DOX is one of the most effective chemotherapy drugs, but the low tumor selectivity and side effects such as cardiotoxicity greatly affected the curative effect [60]. Its conjugation with hydrophilic polymers and an integrin targeting group could be an attractive approach to deliver insoluble and highly toxic drugs to tumor sites and overcome the side-effects.

Li et al. [61] realized a polymer–dual-drug conjugate that could selectively release the two cytotoxic agents—doxorubicin (DOX) and bortezomib (BTZ)—in the intracellular endosomes by means of an acidic condition. An oxidized dextran (Dex- CHO) was functionalized by Schiff base and boronic esterification reactions with cyclo-(Arg-Gly-Asp-D-Phe-Lys) (c(RGDfK)), doxorubicin, and bortezomib. The thus obtained dual-drug conjugate could be used for a synergic cancer therapy: doxorubicin is an inhibitor of topoisomerase II whereas BTZ inhibits proteasomes (Figure 9). The cyclic RGD peptide would be able to mediate a selective active transport of the conjugate to target α_v_β_3_ integrins, overexpressed on cancer cell. The acidic endosomal pH could be able to induce the release of DOX and BTZ from the molecular cargo through the degradation of Schiff base and borate bonds. Then, a simultaneous inhibition of topoisomerase II and NFkB and cell proliferation in vitro and in vivo was realized. The internalization and subcellular localization of DOX were detected by confocal laser scanning microscopy. The in vitro cytotoxicity of polysaccharide–dual-drug cargo toward B16F10 cells were detected by MTT assay. The dual-drug-targeted conjugate Dex-g-(DOXþBTZ)/cRGD showed better inhibition efficacy in proliferation tests compared with the dual-drugs cargo Dex-g-(DOXþBTZ) as a benefit of the cRGD mediated active cell internalization. The ex vivo DOX fluorescence intensities of main organs from euthanized mice was evaluated and showed the highest concentration of DOX in the tumor. This result should be ascribed to the RGD-targeted tumor cell recognition and internalization, and accounted for an effective DOX release in the tumor cells. Study of the biodistribution indicated that the conjugation with the dextran decreased the accumulation of DOX in the liver, thereby with a reduced toxicity. The highest average signal of DOX in the tumor appeared in the Dex-g-(DOXþBTZ)/cRGD group, and it was 10.7 and 3.4 times higher than those of free DOX and the non-targeted Dex-g-(DOXþBTZ), respectively. The Dex-g-(DOXþBTZ)/cRGD also showed good results in vivo tests in decreasing the tumor growth. The results demonstrated that the targeted dual-drugs cargo Dex-g-(DOXþBTZ)/cRGD exhibited an enhanced tumor-targeted action, and this might have great potential in synergistic chemotherapy of cancers.

Mansur et al. [62] conjugated doxorubicin by amide bond with tripeptide (RGD) and L-arginine (Arg) as cell-penetrating amino acid for a synergistic targeting and enhancing internalization by cancer cells (Figure 10).

Carboxymethyl cellulose (CMC) polysaccharide was selected as the biocompatible water-soluble polymer backbone carrier. Cell viability MTT assay with the polymer–drug conjugates was evaluated toward three types of cancer cells (osteosarcoma, SAOS; glioblastoma, U-87 MG; breast cancer, MCF7) and normal cells (reference HEK 293T). The DOX-conjugate showed a lower toxicity to normal cell and a good selective toxicity against cancer cells.

Feni et al. [63] tried to combine the power of cell-penetrating peptides (CPPs) [64] with the selectivity of an integrin receptor ligand to develop a peptide–drug conjugate with doxorubicin. The integrin-targeting unit (c[DKP-f 3-RGD]) was composed of the RGD tripeptide cyclized within a diketopiperazine and characterized by a high selectivity toward integrin α_v_β_3_ (Figure 11).

Binding affinity of the new conjugates to the isolated integrins α_v_β_3_ and α_v_β_5_ was evaluated and with low nanomolar affinity, indicating that the presence of the CPP did not interfere with the integrin-targeting portion. All DOX-loaded conjugates showed high toxicity in U87 glioblastoma, MCF-7 breast cancer, and HT-29 colon carcinoma cells with EC50 in the range 8 to 53 μM.

Tripodi et al. [65] developed a Daunorubicin (Dau) conjugate with cyclic asparagine–glycine–arginine (NGR) peptide to target integrins (Figure 12). The in vivo studies on KS bearing mice the NGR-Dau conjugates did not cause toxic side effects to the animals during the treatment in comparison with Dau alone. Tumor growth inhibition of Kaposi Sarcoma by conjugate NGR-Dau was higher in comparison with Dau. Non-significant liver/body weight ratio changes could be detected in NGR-Dau conjugates respect the free Dau treated groups proving evidences for selectivity and non-toxicity to healthy tissue of the NGR-Dau conjugates.

Liang et al. [66] realized three DOX conjugates with different linkers to the cRGD portion (Figure 13). The new conjugate RSDOX showed a stimulus-triggered drug release and lower toxicity than free DOX. The cRGD fragment was responsible of an enhanced cellular uptake toward the α_v_β_3_-expressing B16 cells via a receptor-mediated endocytosis, and a consequent higher intracellular DOX concentration. In vitro tests showed a lower cytotoxicity of conjugates than free drug. Whereas in vivo testing evaluated on a mouse model of B16 tumor-bearing C57BL/6, showed that the conjugates exhibited a significant inhibition of tumor growth 1.4–1.7-fold of free DOX. This enhanced effect should be likely attributed to the combination of α_v_β_3_ mediated active targeting and the linker that trigger an efficient intracellular drug release. RSDOX was localized in lysosomes, where it can deliver more DOX to cytoplasm by amide hydrolysis thus leading to a significant inhibition of tumor growth and minor side effects. Indeed no significant body weight loss was observed in the treated group, thus suggesting a low systemic toxicity.

## 5. Multi-Kinase Inhibitors

### Sunitinib

Sunitinib is a tyrosine kinase inhibitor and a standard of care in the treatment of metastatic renal cell carcinoma. It inhibits the vascular endothelial growth factor receptor (VEGFR) and other tyrosine kinases, including the platelet-derived growth factor (PDGF), and c-kit receptor at nanomolar concentrations. Sunitinib is generally well tolerated; however, it is associated with adverse effects that can impact quality of life and adherence to therapy [67].

Bianchini et al. [68] designed a sunitinib-conjugate with a cyclic RGD portion that could selectively direct the conjugates toward α_V_β_3_-integrin overexpressing cells, while the sunitinib portion would have exerted its anti-angiogenic properties (Figure 14). Integrin receptor α_V_β_3_ represents an eligible target for the selective discrimination of cancer cells due to its overexpression in advanced melanoma cells and its recognized role in metastatic disease progression [20]. On tumor-bearing mice, compound A and B reduced the growth of melanoma xenografts compared to free sunitinib at low doses, and selectively localized in tumor tissue. These conjugates showed a selective uptake by melanoma cells mainly due to their binding ability to the α_V_β_3_-integrin, the observed in vitro and in vivo selectivity towards melanoma would be a good premise for the effectiveness of a targeted therapy [69,70].

## 6. Camptothecin

Utilizing a short non-RGD cyclic peptide, ALOS4, previously studied to allosterically bind to integrin α_v_β_3_, Yacobovich et al. [71] conjugate the anticancer drug Camptothecin (CPT) to ALOS4 to achieve increased chemo-stability of CPT as well as specific internalization of CPT into human malignant melanoma cells to successfully induce DNA damage and cell death (Figure 15). They observed that ALOS4-CPT binding to integrin α_v_β_3_ enables its internalization followed by nuclear accumulation of CPT that results in DNA damage and induction of cell death [72].

Gilad, Y., et al. [73] prepared novel peptide–camptothecin conjugates (Figure 16). A selective cytotoxicity of two representatives—one linear and one cyclic RGD—camptothecin conjugates were evaluated on α_v_β_3_ integrin overexpressed cancer cell lines: H-1299 (human non-small cell lung carcinoma), PC-3 (human prostate cancer), and HEK-293 as a negative control. The cyclic RGD-CPT conjugate showed good potency against H-1299 and PC-3 tumor cells, but a significant toxicity reduction towards the non-tumor HEK cells. Moreover, the conjugate showed a higher percentage of growth inhibition in PC-3 cells than the free camptothecin, 50% vs. 40% at 10 μM, respectively. This tendency was also observed in H1299 cells, but only at concentrations higher than 50 μM.

Dal Pozzo et al. [74] evaluated the camptothecin conjugate depicted in (Figure 17). Hydrazone conjugates exhibited in vitro tumor cell inhibition similar to the parent drug camptothecin.

## 7. Nonsteroidal Anti-Inflammatory Drugs (NSAIDs)

Some cyclooxygenase (COX) inhibitors such as non-steroidal anti-inflammatory drugs (NSAIDs) have shown anticancer properties in in vitro and in vivo studies [75]. Their anticancer properties seemed to occur through COX inhibition, which in turn inhibits α_V_β_3_ activity and suppresses angiogenesis [76]. COX-2 is more critically involved in cancer progression than COX-1. COX-2 expression is induced by inflammatory cytokines and cellular transformation, and its overexpression occurs with many human cancers, including colon, breast, prostate, and skin [77]. Thus, considering the absence of COX-2 from most normal tissues and its overexpression in cancer cells, targeted delivery of these therapeutic agents to the cancerous cells could result in maximizing therapeutic effect and in minimizing side effects [78]. NSAIDs can in fact be selectively delivered into cancer cells, which overexpress α_V_β_3_ integrin, through conjugation to RGD sequence [79].

Shokri et al. [78] synthesized different RGD-conjugates with Naproxen and Ibuprofen (Figure 18) by Fmoc-peptide synthetic strategy [80,81]. Antiproliferative activity was evaluated, and both conjugates did not show significantly improved activity against A2780 (human ovarian carcinoma cell line with normal expression of RGD-binding integrins) and OVCAR3 (as cell line with overexpression of RGD-binding integrins). Therefore, it could indicate that the RGD motif is not qualified as a targeting tool for ibuprofen and naproxen. Nevertheless, the relevant selectivity of the conjugated compounds was verified since no inhibitory activity was observed on MCF-7, a human breast cancer cell line without overexpression RGD binding integrins or fibroblasts as normal non-tumor cells [78].

Mohammadi et al. [75] evaluated the selective delivery of Ketoprofen and Naproxen to tumor cells by conjugation with a RGD-carrier radiolabeled with technetium-99m [^99m^Tc] to trace the delivery of NSAIDs into tumors (Figure 19). In this conjugate, the peptide sequence GAGG was added as chelator ligand to form stable complex with ^99m^Tc. The radiolabeled compounds thus realized showed higher affinity to A2780 cells, which have an overexpression of α_V_β_3_ integrin compared with OVCAR-3 cells. The inhibition of cell proliferation by RGD conjugates was enhanced respect Ketoprofen and Naproxen alone, and comparable with that of Doxorubicin on OVCAR-3 and A2780 cells. The cytotoxicity outcomes probably indicate a synergetic effect of RGD (anti-integrin) and Ketoprofen/Naproxen (anti-proliferation), thus allowing the possibility of fewer side effects in chemotherapy. The selective delivery of RGD conjugates to cells overexpressing α_V_β_3_ integrin was demonstrated by their internalization in radioactivity experiments even if the percentage of internalization was quite low, as predicted for monomeric linear RGD peptides.

## 8. Antimitotic Drugs

### 8.1. Paclitaxel

Antimitotic drugs inhibit polymerization dynamics of microtubules and can be divided into two subgroups according to their mechanism of action: microtubule-destabilizing or microtubule-stabilizing agents. The first ones inhibit the polymerization of microtubules while the second ones, on stabilizing microtubules enhance microtubule polymerization and prevent Ca^2+^ or cold-induced depolymerization, and subsequent disassembly [82]. Specifically, Paclitaxel (PTX) belongs to the second group and was approved by the Food and Drug Administration for the treatment of ovarian, breast, and lung cancer, as well as Kaposi’s sarcoma [83]. However, it also has some limitations such as having extremely poor water solubility and needing a relatively higher dose to take effect compared to other anticancer drugs [84].

Dias et al. [85] synthesized a peptidomimetic integrin ligand cyclo-(DKP-RGD), above described in conjugation with doxorubicin [63], and now conjugated to PTX via a self-immolative spacer, the Asn-Pro-Val (NPV) linker, and a hydrophilic PEG4 spacer (Figure 20). The conjugate showed an activity at nanomolar level toward α_v_β_3_. The mechanism of action provided that the integrin ligand could drive the conjugate accumulation at the tumor site, although it was not so efficiently internalized in α_v_β_3_-expressing cancer cells. Proinflammatory stimuli promoted the release of elastase, responsible of the enzymatic cleavage of the NPV linker and of the release of PTX. NPV tripeptide is in fact a substrate for hydrolysis by neutrophil elastase, whose expression and activity are upregulated in numerous cancer types [86]. The cleavage of the tripeptide and the subsequent release of PTX were verified in the presence of neutrophil elastase, whereas the use of an inactivated enzyme did not lead to any PTX release. The selectivity of linker cleavage was evaluated by treating with a rat liver-derived lysosome extract, composed of a mixture of proteolytic enzymes. Upon 2 h exposure, the conjugate was digested only partially, indicating the possible presence of elastase in the lysosome extract. The antiproliferative activity of cyclo(DKP-RGD)-NPV-PTX against human renal cell carcinoma 786-O, in the presence or absence of elastase was then determined, evaluating the extracellular cleavage of the NPV linker followed by the PTX internalization into cancer. In the absence of elastase, the conjugate did not exhibit a significant cytotoxic activity, whereas free PTX inhibited cell proliferation at nanomolar level. In presence of elastase instead, the conjugate demonstrated an antiproliferative activity higher than the one of PTX (29.5 ± 7.6 nM) in the same conditions [85].

### 8.2. Cryptophycins

Others antimitotic agents widely used for the development of integrin-targeted drug delivery system belong to Cryptophycin’s family, consisting of 16-membered highly cytotoxic macrocyclic depsipeptides, first isolated from cyanobacteria. Their mechanism of action is based on their ability to bind to tubulin, inhibiting microtubule polymerization and depolymerizing preformed microtubules in vitro [87]. Initially, a Cryptophycin-52 synthetic analog was designed but, unfortunately, due to a lack of efficacy in vivo combined to the high neurotoxicity, it failed as potential clinical candidate [88]. A further investigation of structural moieties necessary for biological activity permitted to develop large number of synthetic analogues [89,90,91] that showed excellent antitumor activity (picomolar level) against mammary, colon, and pancreatic adenocarcinomas in mouse xenographs.

Borbély et al. [92] developed and synthetized an RGD–cryptophycin conjugate (Figure 21), consisting of the highly cytotoxic payload, cryptophycin-55 glycinate, and as targeting vehicle the c(RGDfK) peptide. These two portions are connected through the protease-cleavable Val-Cit dipeptide that displays an excellent balance between high stability in circulation and rapid intracellular cleavage in the presence of cathepsin B and other cysteine cathepsins. In order to achieve an efficient release of the cytotoxic agent, the Gly-Pro dipeptide unit was inserted between the drug and the cleavage site. It was designed to decompose by diketopiperazine formation [93]. Moreover, to improve water solubility of the conjugate, a polyethylene glycol (PEG5) spacer was introduced between the integrin ligand and the cleavage site [92].

Integrin binding affinity and in vitro cytotoxicity of the conjugates have been evaluated. The Val-Cit-Gly-Pro peptide linker undergoes enzymatic cleavage resulting in the release of the Gly-Pro-Cry-55gly metabolite, indicating the lack of an efficient self-immolation step. Despite the inefficient release of the free drug, conjugate showed a good potency. Unfortunately, the conjugate showed a poor selectivity for cell lines with different integrin α_v_β_3_ expression, such as M21 (α_v_β_3_+) and M21-L (α_v_−, α_v_β_3_−). The results showed that the RGD–Cryptophycin conjugate was internalized by a nonspecific process attributed both to the RGD ligand and the high hydrophobicity of the payload and/or conjugates [92].

Subsequently, Borbély et al. [94] employed Criyptophycin-55 glycinate for the conjugation with cyclo[DKP-RGD] peptidomimetic for integrin targeted delivery (Figure 22).

In order to connect the integrin ligands to the cytotoxic agent, PEG4 and the cathepsin B sensitive peptide linker (Val-Ala) combined with the *p*aminobenzyl-carbamate (PABC) as a self-immolative spacer. In viability assays, the Cryptophycin-conjugate displayed high potency in vitro, but lower than Cryptophycin-55 glycinate. Moreover, the conjugate showed similar cytotoxic activity on the antigen-positive and antigen-negative cell line, highlighting no correlation between the in vitro antitumor activity of the conjugate and the α_v_β_3_ integrin expression level [94].

Borbély et al. [95] developed a multimeric system to increase the selectivity and binding affinity of the RGD ligands towards integrin α_v_β_3_. In particular, a regioselectively addressable functionalized template (RAFT) cyclodecapeptide scaffold linked to four copies of the functionalized cyclopentapeptide c(RGDfK) is used as vehicle for the delivery of Cryptophicin-55-glycinate. The labeled tetrameric compound RAFT-c(RGDfK)_4_-Cy5 in fact displays a 10-fold higher binding affinity towards isolated integrin α_v_β_3_ compared to the monomeric analog and can be efficiently internalized through the clathrin-mediated endocytic pathway [96]. Furthermore, a cleavable linker was introduced between the ligand and the Cryptophicin-55-glycinate consisting of a PEG5-chain, the protease sensitive Val-Cit dipeptide, and the para-aminobenzyloxy carbonyl (PABC) self-immolative moiety. (Figure 23) The antiproliferative activity of the conjugate was evaluated using three cell lines expressing different levels of integrin α_v_β_3_ (U87 (α_v_β_3_+), M21 (α_v_β_3_+), and M21-L (α_v_−, α_v_β_3_−)). Tetrameric RGD-cryptophycin conjugate exhibit IC_50_ values in the nanomolar range towards the integrin positive cells. In contrast, incubation of the M21-L cells with the conjugate, resulted in marginal cell growth inhibition, demonstrating a great tumor selectivity [95].

### 8.3. Monomethyl Auristatin E (MMAE) and F (MMAF)

Monomethyl auristatin E (MMAE) and F (MMAF) are cytotoxic agents that bind to microtubules and prevent cell proliferation by inhibiting mitosis (Figure 24) [97]. Moreover, they are synthetic analogs of Dolastatin 10, extracted from the sea hare *Dolabella Auricularia*, but they maintain the same so potent cytotoxic activity [98,99] that they cannot be used as drug themselves [100]; they are in fact used as payloads in a number of state-of-the-art antibody–drug conjugates (ADCs) [101,102,103,104,105]. Targeted MMAE and MMAF have different cell killing mechanisms in vivo depending on potencies of the respective drugs, together with the enhanced retention of MMAF, compared to the more lipophilic MMAE, within the tumor cells thanks to the negative charge on the C-terminal phenylalanine residue [106]. The potential of these two cytotoxic agents was also exploited in small molecule–drug conjugates.

Dias et al. [100] conjugated MMAE to cyclo[DKP-isoDGR] using both cleavable and uncleavable linkers, as reported in Figure 25. The antiproliferative activity was tested against human glioblastoma (U87) and human melanoma (M21) cells overexpressing α_v_β_3_ integrins. These tests confirmed that the free drug demonstrated higher antiproliferative activity compared to the conjugates, which is consistent with their inefficient integrin-mediated internalization. Regarding instead the differences between linkers, as expected, conjugate with cleavable linker has higher antiproliferative activity than conjugate with the uncleavable one.

Rivas et al. [107] conjugated to MMAE to cyclo[DKP-RGD] through a β-glucuronidase-responsive linker. Previously reported studies demonstrated that this ligand does not promote a significant receptor-mediated internalization [87], and this behavior can be exploited to develop non-internalizing conjugates in which the release of the drug can be provided by β-glucuronidase, widely present in lysosomes and in tumor extracellular areas [108].

The two MMAE conjugates in Figure 26 differ for the spacers: in compound **A** there is a glutaric acid derivative, while in compound **B** a triazole + PEG4. This aspect could be very important because the linker can affect several characteristics of the conjugates, such as flexibility, solubility, ligand binding affinity, and drug release [109]. Cytotoxicity assays have been conducted by incubating the RGD–MMAE conjugates with α_v_β_3_-expressing cancer cells (87MG human glioma cell line and renal cell carcinoma 786-O cells), in the presence or absence of β-glucuronidase. In the absence of β-glucuronidase, conjugates **A** and **B** revealed a low cytotoxicity compared to MMAE alone. However, in both cell lines the activity of conjugate **B** was found to be a fraction of that of analog **A**, indicating that under these conditions it can be poorly internalized, leading to drug release mediated by lysosomal β-glucuronidase. In the presence of β-glucuronidase instead, the activity of **B** in both cell lines, undergoes an incredibly increase, with IC_50_ values in the same low nanomolar range of free MMAE. No enhancement for compound **A** was observed; it can depend on its structural characteristics and particularly on the spacer used and thus on the drug release [107].

## 9. Others

### 9.1. Benzylguanidine

A way of controlling neural tumors growth, such as in neuroblastoma, could be to control the angiogenesis. It was demonstrated, in fact, that various neuroblastoma cells and angiogenic endothelial cells highly express the integrin α_v_β_3_ receptor [110]. In particular, neuroblastoma (NB) is one of the frequently observed malignant solid tumors of childhood and infancy, accounting for 15% of pediatric cancer deaths [111]. Neural crest tumors often show high norepinephrine transporter (NET) expression. Molecular imaging and treatment of these tumors can be thus also done using radiolabeled norepinephrine analogues, such as *m*-iodobenzylguanidine (MIBG). Nonradioactive BG alone exhibits no anticancer activity, but it can be conjugated to appropriate anticancer drugs, enhancing their selectivity [112]. Karakus et al. [112] conjugated BG to a thyro-integrin α_v_β_3_ antagonist called triazole tetraiodothyroacetic acid (TAT) that showed antiangiogenic activity. In this way, a dual-targeting ligand, recognizing both the NET function and the thyrointegrin receptor, was obtained for the treatment of NB. In particular, TAT, was linked to BG via the polymer linker poly(ethylene glycol)(PEG_400_) (Figure 27). In terms of tumor growth and viability, the conjugate showed an 80–100% increase in neuroblastoma anticancer activity of the antagonist versus control. The in vitro studies of tumor cell targeting also showed that BG-P-TAT had a high affinity for cancer cell binding without any cellular nuclear uptake and that significantly inhibited cell proliferation (about 50–60%) with a concentration-dependent inhibition. The in vivo anticancer efficacy of BG-P-TAT was evaluated in comparison with BG, TAT, and their combination (BG + TAT). They were administered to nude mice xenograft implanted with neuroblastoma. While BG, TAT, and BG + TAT demonstrated 40–50% tumor shrinkage and 40–50% suppression of cancer cell viability, BG-P-TAT showed >80% shrinkage with maximal loss of cancer cell viability.

### 9.2. Dihydrolipoamide Dehydrogenase (DLDH)

A potential cancer treatment is the reactive oxygen species (ROS)-mediated anticancer therapy that shows the advantages of tumor specificity, high curative effect, and less toxic side-effects [113]. It was recognized that, compared with normal cells, many types of cancer cells have high levels of ROS, but an excessive increase can be toxic and can promote apoptotic cell death. Therefore, manipulating ROS levels by redox modulation could be a valuable strategy to selectively kill cancer cells without affecting normal cells [114]. Regarding this, Dihydrolipoamide dehydrogenase (DLDH), a mitochondrial enzyme that comprises an essential component of the pyruvate dehydrogenase complex, could be used as anticancer drug thanks to its ability to produce ROS [115]. Dayan et al. [116] functionalized DLDH enzyme with RGD moieties at the N- and C-termini, in order to target α_v_β_3_ integrins overexpressed on tumor cells. This approach has been used in photodynamic therapy, targeting titanium dioxide nanoparticles complexed with DLDH^RGD^ toward integrin-rich cancer cells; then, Dayan et al. [117] described the use of DLDH^RGD^ for targeted integrin-assisted drug delivery in melanoma cells. The results showed that modification of DLDH did not affect its enzymatic activity and kinetic parameter, suggesting that DLDH^RGD^ retained its structural conformation. Then, DLDH^RGD^ α_v_β_3_-dependent intracellular penetration was confirmed in both normal and melanoma cells. Moreover, the fast uptake of DLDH^RGD^ into the cells led to apoptosis and reduction in cell number in mice and human melanoma, while integrin-positive normal cells remained intact, indicating that DLDH^RGD^ is a potent ROS generator also in living cells. The efficacy of DLDH^RGD^ in a B16F10 mice melanoma model, which presenting rapid tumor growth and metastasis along all inoculation routes, inhibited tumor growth as well as metastases [117].

## 10. Conclusions and Perspectives

Integrins have been gained an increasing interest as target receptors for the internalization of covalent conjugates between cytotoxic drugs and integrin ligands. This specific internalization pathway could have a significant role in determining efficacy and selectivity of the drug conjugates. Several compounds showed selective toxicities against cancer cells respect to normal cells and this result is of good omen in diminishing side-effects of chemotherapy.

Notwithstanding the good results obtained by the new conjugates, in our opinion some points are still open for future developments. Here are some suggestions:In order to select more specific integrin ligands, much more attention should be dedicated in a deep investigation on integrin expression/overexpression in cancer cells.RGD or cRGDs are the main peptides employed in these conjugates; however, in the literature much more integrin ligands have been developed, so other molecules, especially those more selective, should be investigated as targeting moieties.Some integrins are going to emerge as interesting targets in cancer cells and could be investigated, such as α_4_β_1_ integrin in multiple myeloma resistant to bortezomib, β_7_ [118,119,120], α_v_β_6_ [121], and α_3_β_1_ integrins [122].Application of selective integrin agonists could enhance a more selective internalization of the conjugates [123].

The research summarized herein could increase interest and provide a more rational approach to develop safer and more effective therapeutics, as a good premise for the effectiveness of targeted therapies.

## Figures and Tables

**Figure 1 cancers-13-00299-f001:**
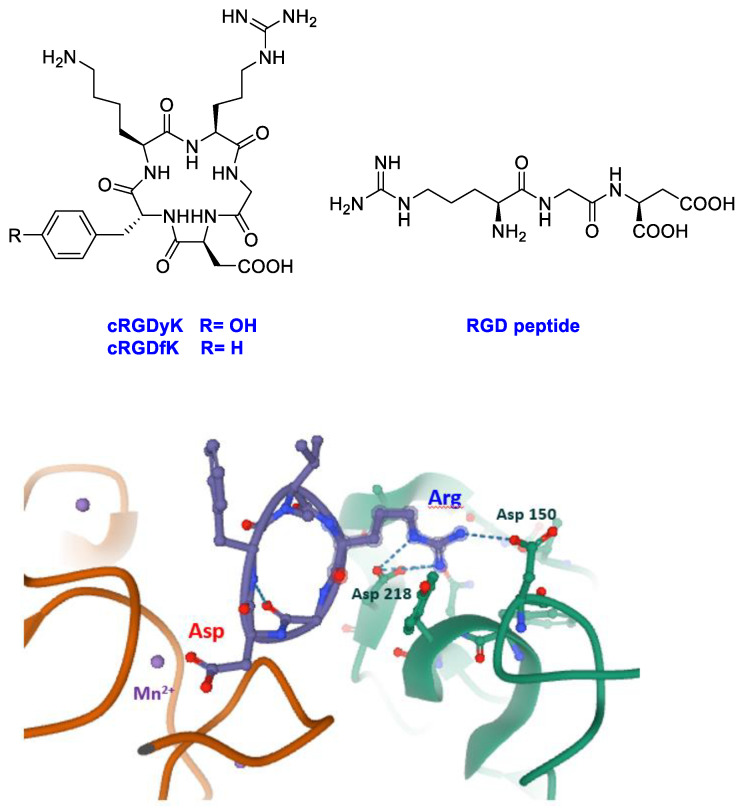
Linear (RGD) and cyclic peptides mostly used to target integrins; snapshot of the ligand-binding region of integrin α_v_β_3_ with cRGDfK (structure 1L5G, PDB databank, α_v_ green chain, β_3_ orange chain, cRGDfK violet).

**Figure 2 cancers-13-00299-f002:**
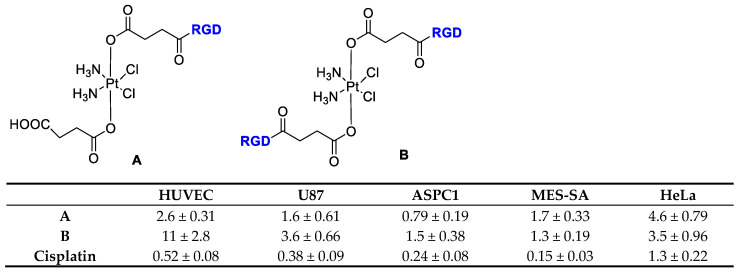
Inhibition of proliferation by Cisplatin and RGD-Targeted Pt(IV). Compounds (**A**) and (**B**) in endothelial and tumor cells, data refer to IC_50_ (μM).

**Figure 3 cancers-13-00299-f003:**
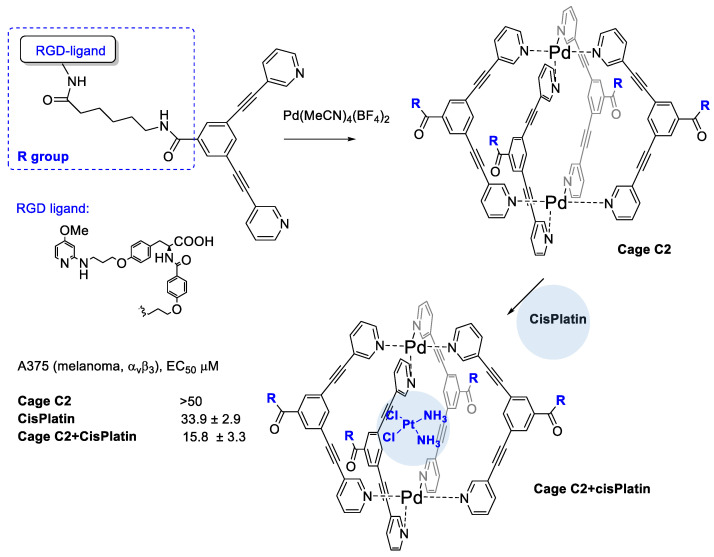
Metallacage decorated with RGD peptides to target integrins and to entrap cisPlatin drug.

**Figure 4 cancers-13-00299-f004:**
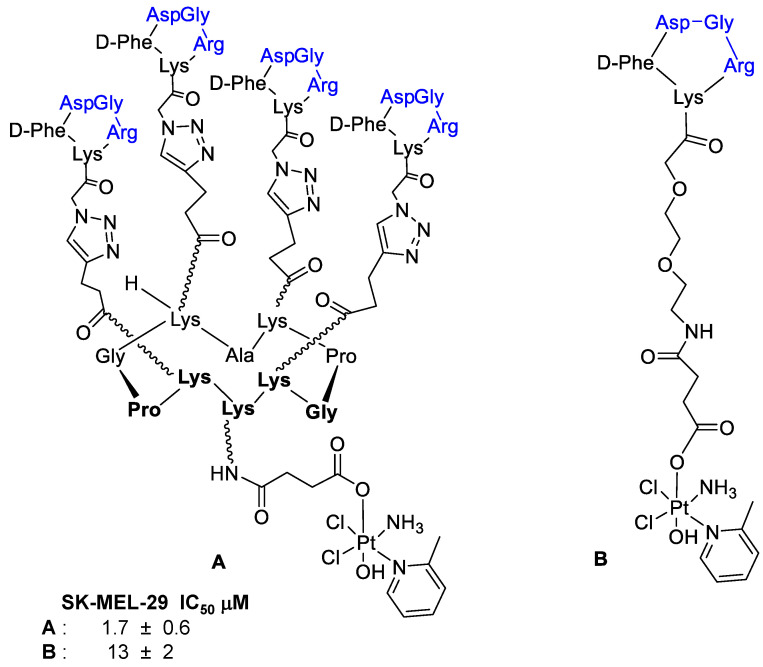
Tetrameric (**A**) and monomeric (**B**) RAFT-RGD systems as platinum (IV) complexes.

**Figure 5 cancers-13-00299-f005:**
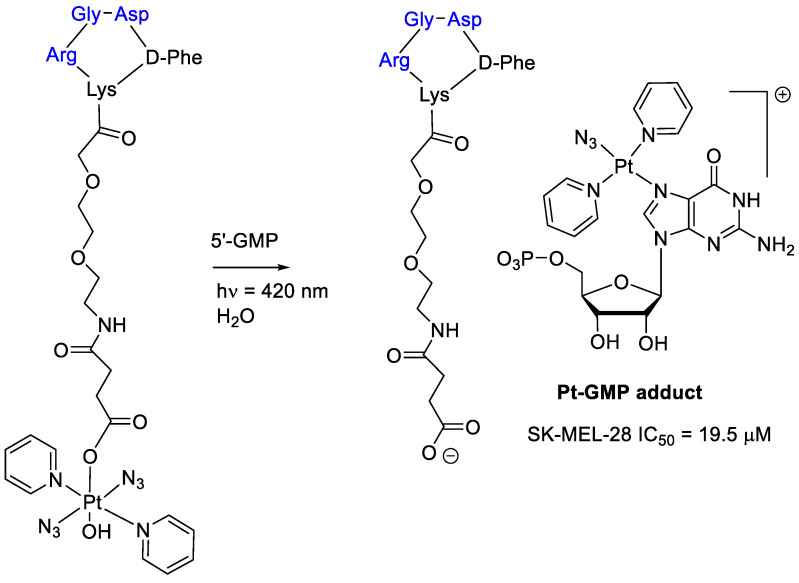
Platinum(IV) prodrug linked to a cyclic RGD–peptide and its guanosine 5-monophosphate adduct obtained by photoactivation.

**Figure 6 cancers-13-00299-f006:**
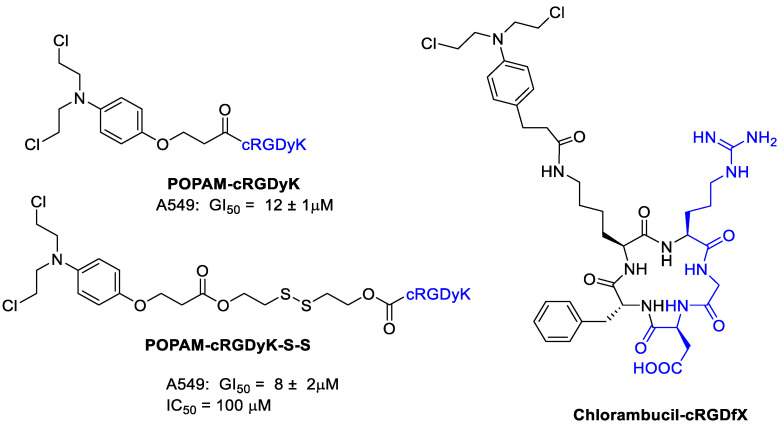
Nitrogen mustard derivatives conjugated with cyclic RGD integrin ligand.

**Figure 7 cancers-13-00299-f007:**
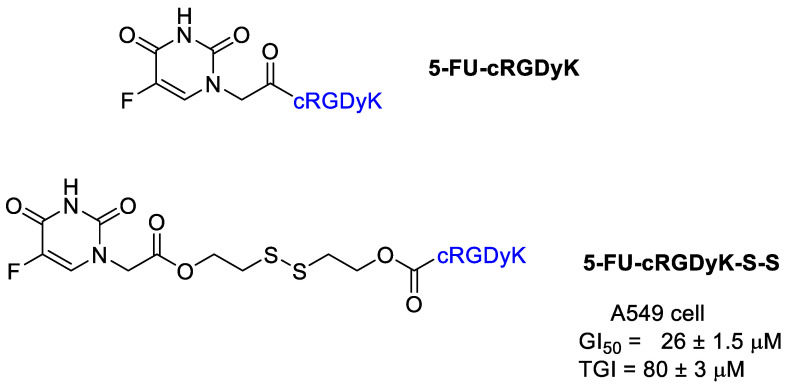
5-Fluorouracil conjugates with cRGDyK.

**Figure 8 cancers-13-00299-f008:**
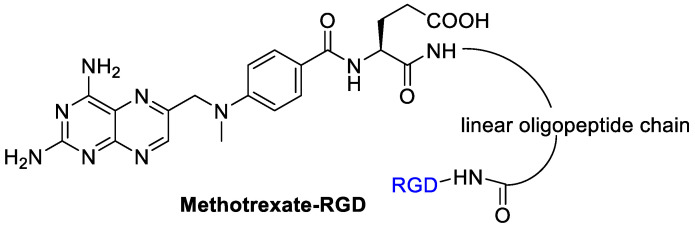
RGD peptide as integrin targeting carrier for methotrexate.

**Figure 9 cancers-13-00299-f009:**
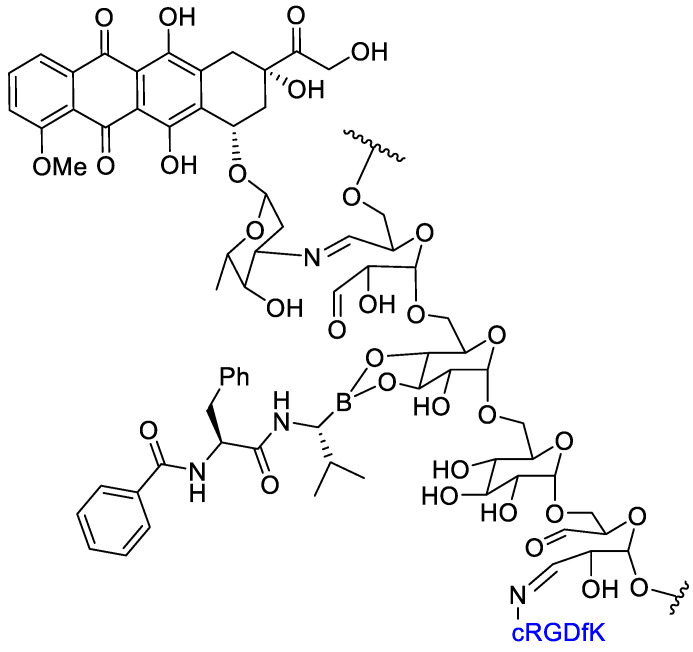
Dual-drug conjugate: doxorubicin and bortezomib on oxidized dextran with cRGD for integrin targeting.

**Figure 10 cancers-13-00299-f010:**
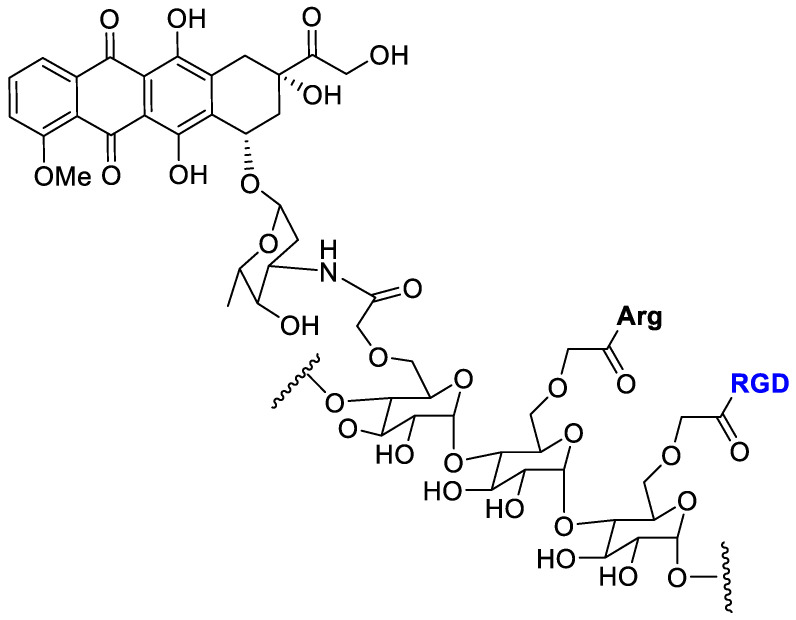
Doxorubicin conjugated with carboxymethylcellulose decorated with arginine and the RGD peptide.

**Figure 11 cancers-13-00299-f011:**
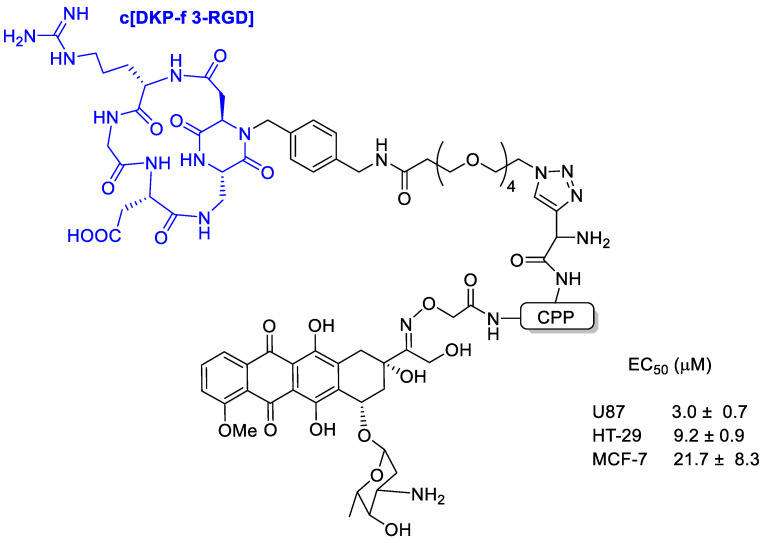
Doxorubicin conjugated with the integrin-targeting unit c[DKP-f 3-RGD] composed of the RGD tripeptide cyclized within a diketopiperazine and characterized by a high selectivity toward integrin α_v_β_3._

**Figure 12 cancers-13-00299-f012:**
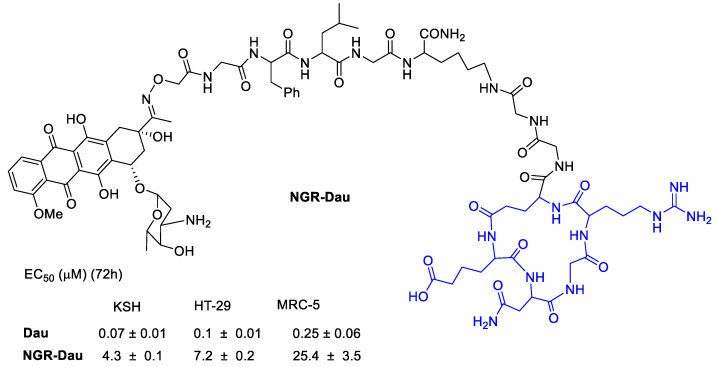
Daunorubicin conjugated by oxime linkage to a polypeptide with a NGR integrin ligand.

**Figure 13 cancers-13-00299-f013:**
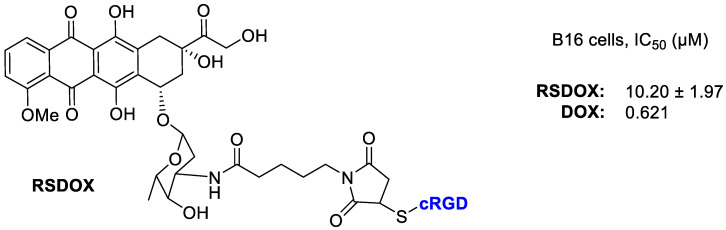
Doxorubicin conjugated with RGD peptide via sulfur link to succinimide.

**Figure 14 cancers-13-00299-f014:**
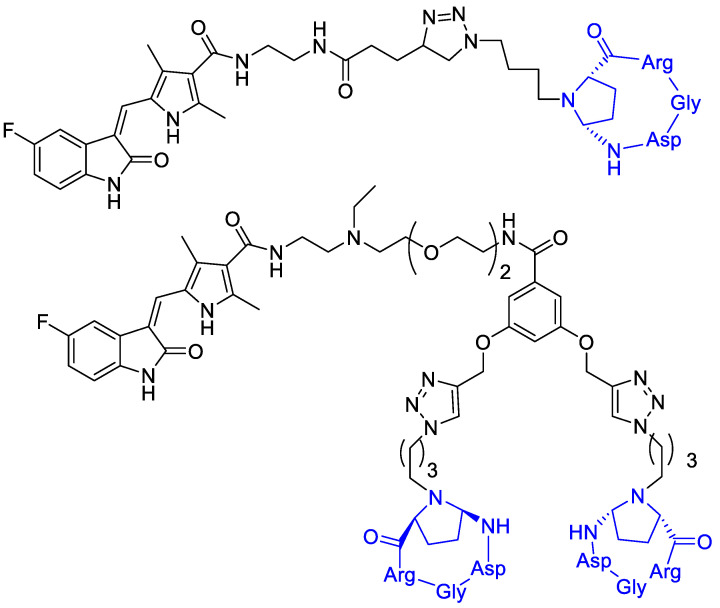
Sunitinib is linked to a cyclic RGD portion that selectively directs the conjugates toward α_V_β_3_-integrin overexpressing cells.

**Figure 15 cancers-13-00299-f015:**
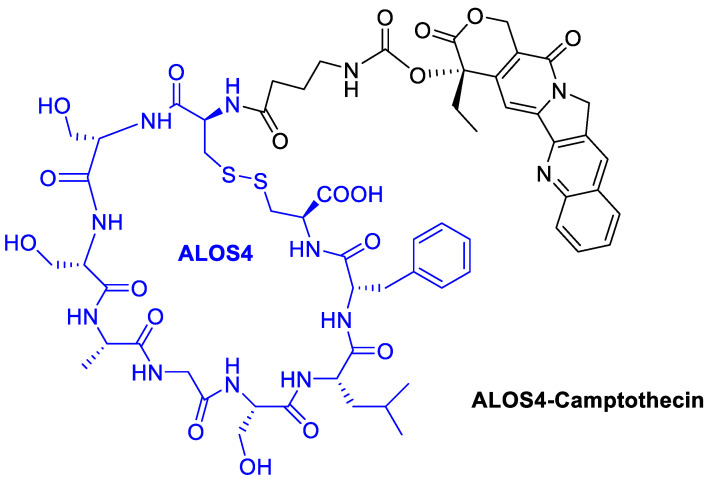
Camptothecin conjugated with the cyclic peptide ALOS4 to bind integrin α_v_β_3_.

**Figure 16 cancers-13-00299-f016:**
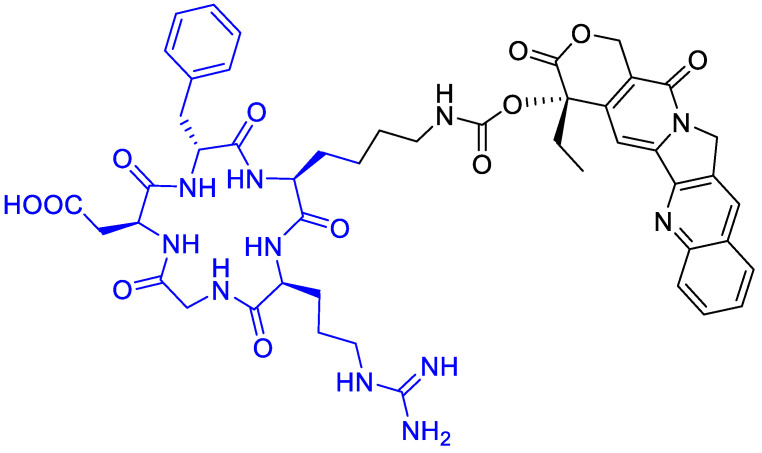
Cyclopentapeptide to carry camptothecin with a short linker.

**Figure 17 cancers-13-00299-f017:**
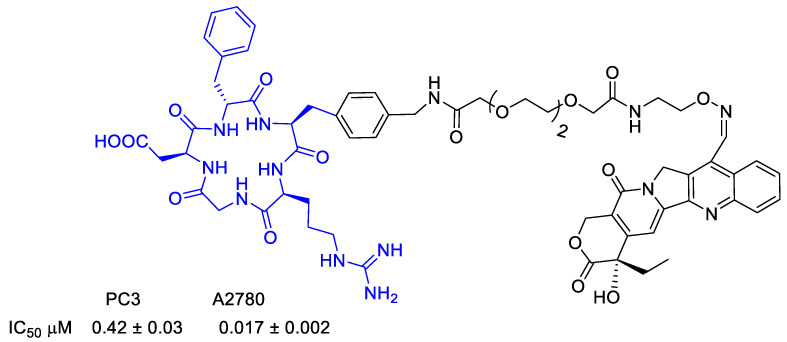
Camptothecin conjugated to a cyclopeptide with a multifunctional linker.

**Figure 18 cancers-13-00299-f018:**
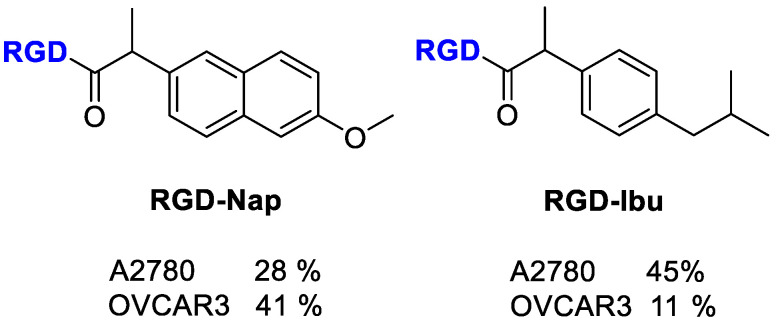
Structure of RGD-conjugates to Naproxen and Ibuprofen; inhibition of cell proliferation (%) by the compounds at 100 µM.

**Figure 19 cancers-13-00299-f019:**
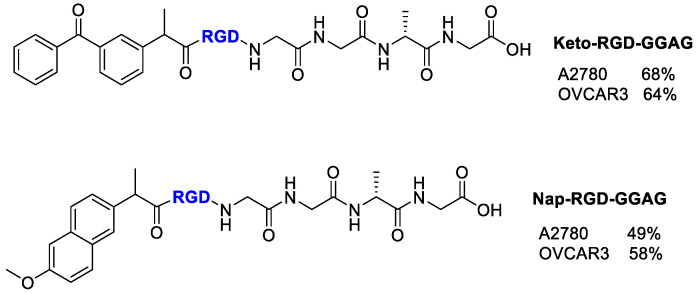
Structure of Ketoprofen-RGD-GGAG peptide, Naproxen-RGD-GGAG peptide and inhibitory % values of conjugates at 100 µM.

**Figure 20 cancers-13-00299-f020:**
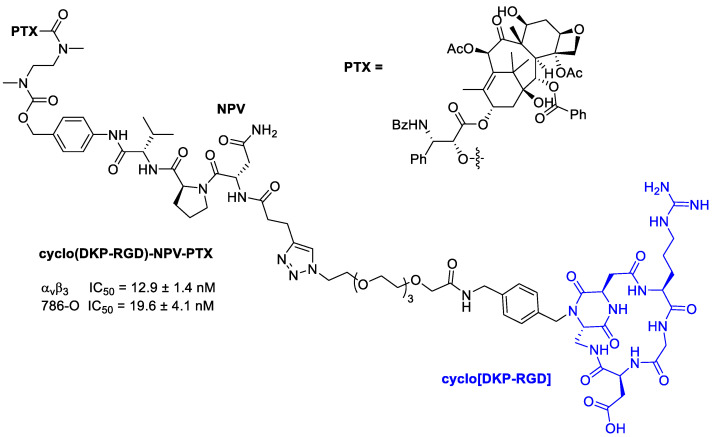
Molecular structures of cyclo(DKP-RGD)-NPV-PTX; IC_50_ of biotinylated vitronectin binding to the α_v_β_3_ receptor; antiproliferative activity of cyclo(DKP-RGD)-NPV-PTX in α_v_β_3_-expressing human renal cell carcinoma 786-O cells in the presence of elastase from human leukocytes.

**Figure 21 cancers-13-00299-f021:**
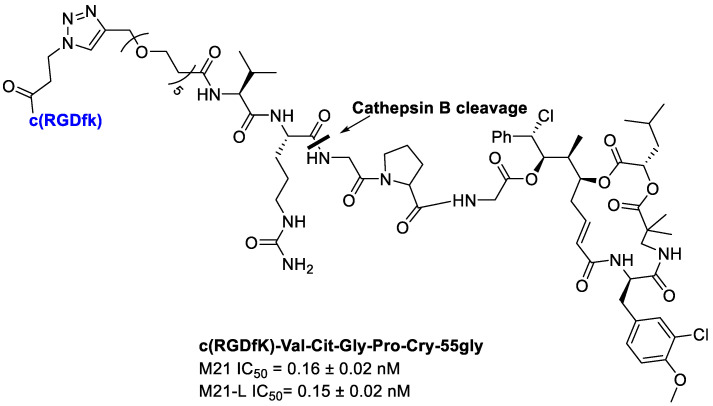
Conjugate c(RGDfK)-Val-Cit-Gly-Pro-Cry-55gly, antiproliferative activity against M21 and M21-L human melanoma cells.

**Figure 22 cancers-13-00299-f022:**
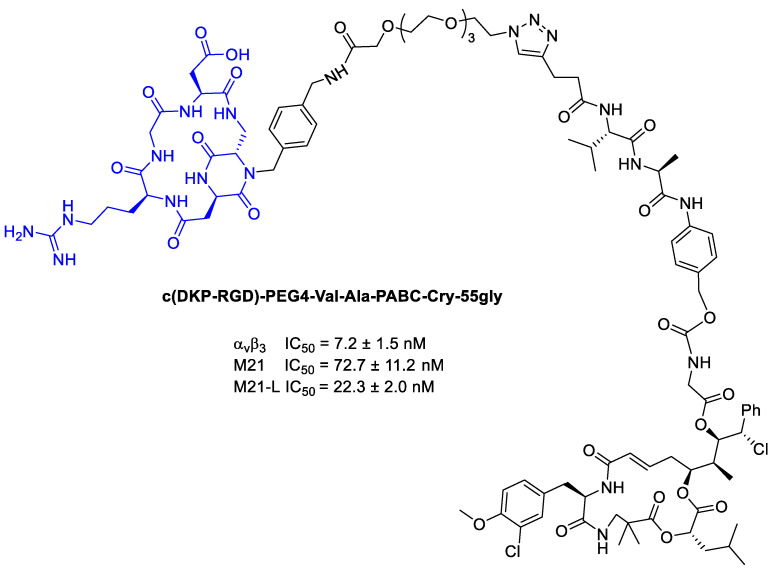
Inhibition data of cyclo[DKP-RGD]-PEG4-ValAla-PABC-Cry-55gly in biotinylated vitronectin binding to human integrin α_v_β_3_ assays, and antiproliferative activity of against M21 and M21-L cell lines.

**Figure 23 cancers-13-00299-f023:**
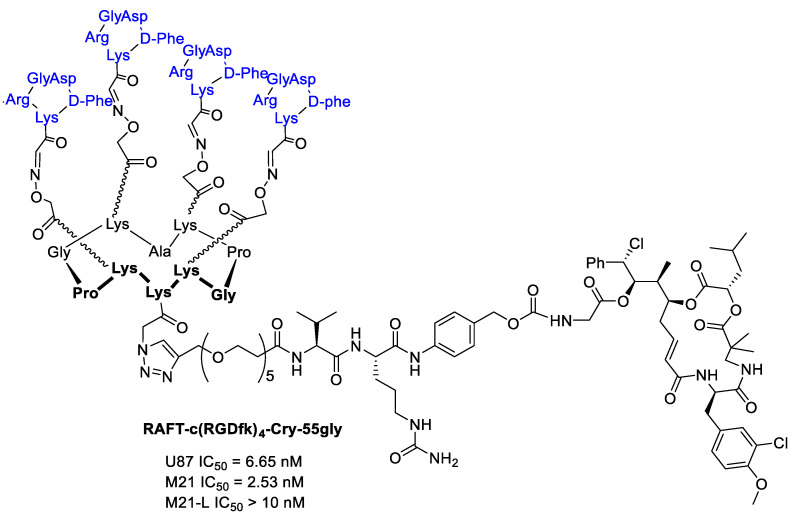
RAFT-c(RGDfk)_4_-Cry-55gly and antiproliferative activity against U87 human glioblastoma, M21 and M21-L human melanoma cell lines.

**Figure 24 cancers-13-00299-f024:**
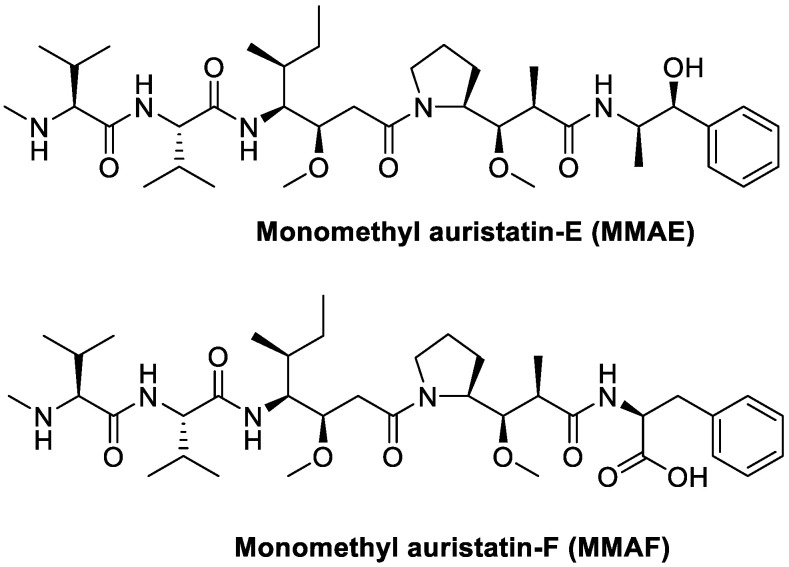
Molecular structures of auristatins MMAE and MMAF.

**Figure 25 cancers-13-00299-f025:**
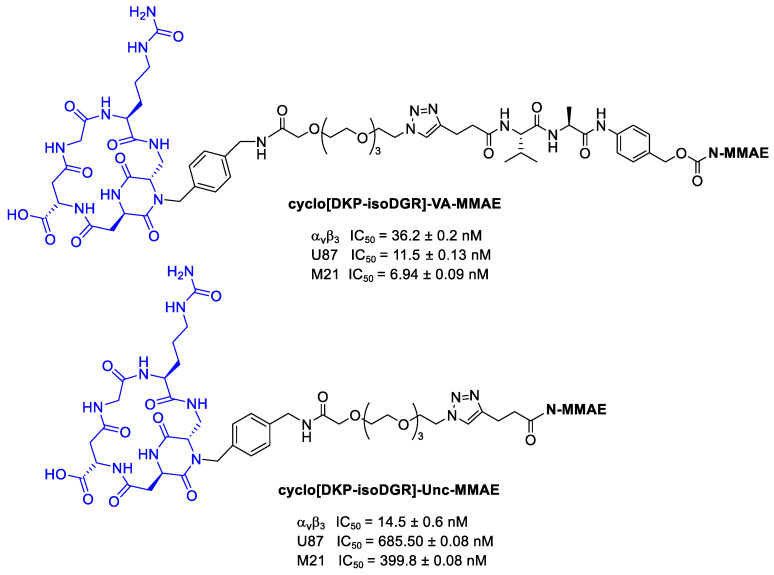
MMAE conjugates: cyclo[DKP-isoDGR]-VA-MMAE and cyclo[DKP-isoDGR]-Unc-MMAE; inhibition of biotinylated vitronectin binding to the isolated α_v_β_3_ receptor; antiproliferative activity of cyclo[DKP-isoDGR]-VA-MMAE and cyclo[DKP-isoDGR]-Unc-MMAE against U87 and M21 cancer cell lines.

**Figure 26 cancers-13-00299-f026:**
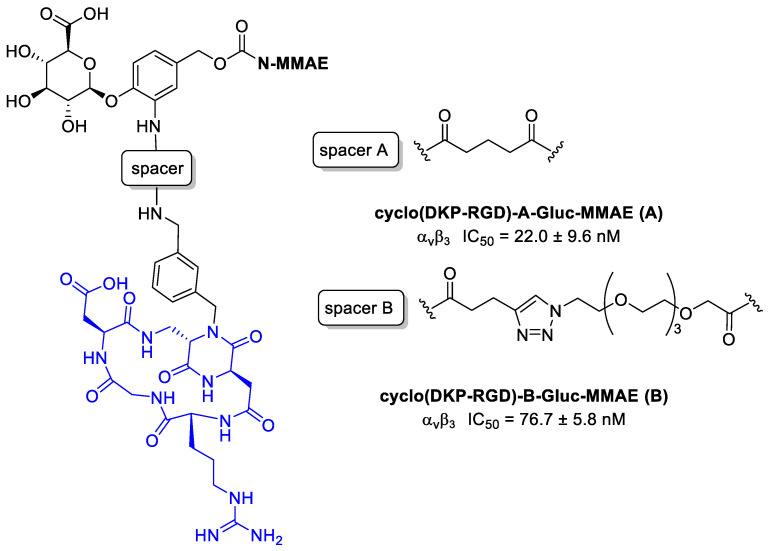
Structure of α_v_β_3_ integrin conjugates cyclo(DKP-RGD)-A-Gluc-MMAE (**A**) and cyclo(DKP-RGD)-B-Gluc-MMAE (**B**); inhibition of biotinylated vitronectin binding to the α_v_β_3_ receptor.

**Figure 27 cancers-13-00299-f027:**
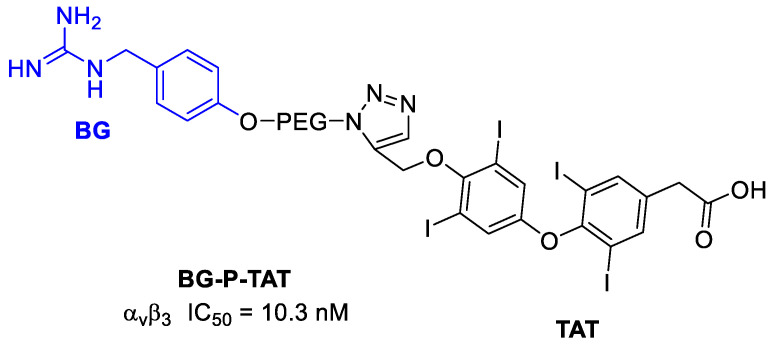
Structure of BG-P-TAT and inhibition potency.

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
