# Peer review of "Molecular Delivery of Cytotoxic Agents via Integrin Activation"

_cancers, 2021, doi:10.3390/cancers13020299_

Round 1

Reviewer 1 Report

This review paper introduces the RGD-conjugation to target the cancer cells by interacting between agents and integrins. Various RGD-modified agents (focused on anticancer drugs), which can help readers understand this technique. This manuscript is well-organized and if some requirements in the following are modified, this paper will be able to provide readers with a better understanding.

  1. Although the interaction between integrin and RGD is a well-known phenomenon, it would be easy to understand if this paper has a figure of the binding of RGD to the site (αv and β3) of integrin.
  2. The RGD-modified anticancer agents, linear and cyclic RGD are employed to develop progressive anticancer drugs. To understand easily for readers, a description of the physicochemical differences between the linear and cyclic structure of RGD should be added.
  3. Some of in vivo and in vitro are not italicized (line 276, 345, and 434). Please use the same format.
  4. Please correct the spelling of 3. Antimethabolites to Antimetabolites.
  5. ± symbol in Figures 6 and 12 should be used in the same format compare to the symbols in other figures.
  6. Please correct the abbreviation of poly(ethylene glycol) on line 561 (PG400 to PEG400).
  7. Integrin αvβ3 on lines 164 and 165 should be written in the same format as others.

Author Response

We are very grateful to this Reviewer for her/his suggestions. To meet these requests, we added a paragraph (lines 63-84) that contains comments on interaction of RGD peptide to integrin and some differences in properties  and the improvement of cyclic peptide over linear ones for the  increased potency as inhibitors. We modified Figure 1 inserting an x-ray structure to explain the interactions of RGD peptide in alph4beta3 integrins and two mor references.

All changes are highlighted

Reviewer 2 Report

This is a robust review which provides an excellent summary of use of covalent conjugates between cytotoxic drugs and integrin ligands as more selective against tumor cells with less side effects than the free drug. It contains and summarizes many different studies on covalent conjugates into a handy and useful resource. The paper covers an interesting topic and it is concise and well written, citing all relevant literature. Therefore, I support its publication.

I only have many minor comments, most of them typos, that need to be addressed:

  1. There is a mixture of citations with and without initials, I listed those with initials which it seems to me unnecessary.
  2. Line 37 “via” should be in italics
  3. Line 43 “via” should be in italics
  4. Line 165 “avb3” should be “αvβ3”
  5. Line 186 delete “acts as”
  6. Line 189 “via” should be in italics
  7. Line 222 delete “D”.
  8. Line 228 insert space between peptide and would in “peptidewould”
  9. Line 250 delete “A.A. P.”
  10. Line 254 delete “of”
  11. Line 261 delete “L.”
  12. Line 275 Define “NGR”
  13. Line 285 delete “Y.”
  14. Line 288 “via” should be in italics
  15. Line 299 “via” should be in italics
  16. Line 308 delete “F.”
  17. Line 225: delete “(human embryonic kidney 293)” because these cells were defined in a text above
  18. Line 276 “in vivo” should be in italics
  19. Line 345 “in vitro” should be in italics
  20. Line 363 define the origin of “A2780” cells
  21. Line 368 define the origin of “MCF-7” cells
  22. Line 403 “via” should be in italics
  23. Line 434 “in vitro” should be in italics
  24. Line 454 “in vitro” should be in italics
  25. Line 482 “internalizes” should be “internalized”
  26. Line 509 “Dias et al” should be “Dias et al.”
  27. Line 522 “Rivas et al” should be “Rivas et al.”
  28. Line 524 “ca” should be “can”
  29. Line 537 define the origin of “87MG” cells
  30. Line 551 “Neuroblastoma” should be “neuroblastoma”
  31. Line 557 “Karakus et al” should be “Karakus et al.”
  32. Line 561 “via” should be in italics
  33. Line 583 and 587 “Dayan et al” should be “Dayan et al.”
  34. From line 583 to 586 there are 3 consecutive references Dayan et al.; Dayan et al. and 114. However, in the reference list there are three references with the same first author Dayan et al. i.e. 113, 114 and 115. It is confusing and not clear which reference is cited. It might be better not to use “Dyan et al.” or to use reference number immediately after mentioning Author et. al. (114). This might be applied in the whole article because reference numbers are always given at the end of the sentence.

Author Response

Thank you very much for the accurate improvements that this reviewer suggested.

 We done all his/her corrections in the list.

All changes are highlighted